## Registered report

psychology

risk communication, fact box, decision aid, decision-making, risk literacy, medical decision

**Author for correspondence:**
Cameron Brick
e-mail: cb954@cam.ac.uk

# Risk communication in tables versus text: a registered report randomized trial on 'fact boxes'

Cameron Brick[1,2], Michelle McDowell[3,4] and Alexandra L. J. Freeman[1]

[1]Winton Centre for Risk and Evidence Communication, Centre for Mathematical Sciences, University of Cambridge, Wilberforce Road, Cambridge, CB3 0WA, UK
[2]Department of Psychology, University of Amsterdam, 1012 WX Amsterdam, The Netherlands
[3]Harding Center for Risk Literacy, University of Potsdam, 14469 Potsdam, Germany
[4]Max Planck Institute for Human Development, 14195 Berlin, Germany

 CB, 0000-0002-7174-8193; MM, 0000-0002-1522-757X;
ALJF, 0000-0002-4115-161X

Objectives: identifying effective summary formats is fundamental to multiple fields including science communication, systematic reviews, evidence-based policy and medical decision-making. This study tested whether table or text-only formats lead to better comprehension of the potential harms and benefits of different options, here in a medical context. Design: pre-registered, longitudinal experiment: between-subjects factorial 2 (message format) × 2 topic (therapeutic or preventative intervention) on comprehension and later recall (CONSORT-SPI 2018). Setting: longitudinal online survey experiment. Participants: 2305 census-matched UK residents recruited through the survey panel firm YouGov. Primary outcome measure: comprehension of harms and benefits and knowledge recall after six weeks. Results: fact boxes—simple tabular messages—led to more comprehension ($d = 0.39$) and slightly more knowledge recall after six weeks ($d = 0.12$) compared to the same information in text. These patterns of results were consistent between the two medical topics and across all levels of objective numeracy and education. Fact boxes were rated as more engaging than text, and there were no differences between formats in treatment decisions, feeling informed or trust. Conclusions: the brief table format of the fact box improved the comprehension of harms and benefits relative to the text-only control. Effective communication supports informed consent and decision-making and brings ethical and practical advantages. Fact boxes and other summary formats may be effective in a wide range of communication contexts.

# 1. Introduction

To make an informed decision, individuals need to understand the potential impacts of their options. However, decision-makers are rarely given a balanced, clear and quantitative summary of the potential outcomes of a decision [1]. Identifying effective summary formats is fundamental to multiple fields including science communication, systematic reviews, evidence-based policy and medical decision-making. Communicators ranging from governments or companies to medical professionals are struggling to provide balanced, thorough, and comprehensible summaries of options. This requires: (i) identifying the most important outcomes, (ii) gathering and summarizing the evidence, and (iii) communicating the evidence such that it is easily comprehended. There is some limited evidence that summary formats such as tables may improve comprehension and short-term recall of the benefits and harms of health interventions [2–4], but the effect and its magnitude are unclear. The studies vary widely in outcomes and the messages have different benefit-to-harm ratios. In this study, we examine effective communication with a promising and simple evidence communication format, the *fact box*, using a high-powered, pre-registered, longitudinal, representative-sample design across two medical topics (preventative and therapeutic).

Much of the previous research on effective communication assumes a correct behaviour such as exercise or treatment adherence and tries to change attitudes or behaviour, e.g. [5]. However, it is rarely clear which choice or option is universally correct. In line with the UK Montgomery ruling [6] and similar laws in Australia and elsewhere, standard practice across many fields is orienting towards the ethical principle of informed choice. Communications that are designed to boost comprehension can support individuals in aligning choices with their own values. Communications that support individuals in choosing based on their own values have legal, ethical and psychological advantages [7]. For example, only patients can know the relative personal importance of quantity versus quality of life in their decision-making, yet communications can aim to ensure that the magnitude of these outcomes is communicated clearly to facilitate their choice. Therefore, it is necessary to identify communication formats that lead to better understanding.

Previous research has examined how to describe potential harms and benefits to effectively inform rather than to persuade [8]. It is best practice to test the effectiveness of communications within the target population for validity [9] and across different populations for generalizability (as in the current design). The risk communication literature provides general principles on how to identify types of uncertainty [10] and represent them visually [11], how to accomplish quality graphical design [12], and how to communicate numbers (e.g. avoid expressing risk in the poorly understood 1/N format; for a review, see [13]. However, many studies evaluating different formats communicate only a few pieces of information (e.g. one harm and one benefit) rather than multiple quantities (e.g. multiple harms and benefits), which is the more realistic use case (for exceptions, see [14,15]). What is lacking is a robust, well-powered study to test the comprehension of numeric benefits and harms in a table versus text only.

The results of the current study will support evidence-based communication in fields where harms and benefits need to be summarized for multiple options. A key area of public need is communicating policy options to decision-makers and voters. A recent review identified four key challenges for communicating the impacts of policy options [16]: broad and heterogeneous effects, outcomes with different metrics, potentially long timescales and large uncertainties. These challenges increase the inherent tension between coverage and comprehensibility in messages. Currently, there is not enough evidence to deliver confident recommendations to policy communicators on how to communicate policy options [17]. This study is focused on individual decision-making (e.g. which treatment one would recommend for a family member) but is designed to allow easy extension in the future to policy decisions (e.g. 'should this treatment be recommended to everyone') if the tabular format is more successful than plain text in communicating potential harms and benefits in this study.

## 1.1. Fact boxes

The fact box is a brief tabular presentation of expected outcomes between interventions based on an earlier tabular format [18]. Fact boxes are designed to be read quickly and be understood even by lower-numeracy individuals [15]. They are a promising evidence communication format with a growing number of evaluation studies and also a start-up company [19]. Fact boxes may lead to greater ease of use and liking compared to text [20]. In particular, information may be easier to locate

and extract and/or be attended to more because of visual features that lead to engagement. More attention and engagement is likely to be helpful; decision latency appears to lead to better decisions [21].

Evidence is emerging that treatment outcomes including side effects can be adequately understood from fact boxes [2–4], but most studies did not use a control with the same content. One paper compared a table, bar chart, risk scale, frequency (flow) diagram, and an icon array, and found that the table and flow diagram were best understood [22]. Another experiment compared fact boxes to control messages and measured comprehension and liking, but the control message was the US Food and Drug Administration drug information leaflets [20], which are long, hard to understand, and do not contain the same harms and benefits information as the fact boxes. Therefore, it is not clear how much the better performance of the fact boxes was affected by differences in content rather than format. A recent paper found that whether icon arrays were included in fact boxes did not affect comprehension, so icon arrays appear to be equivalently effective [14]. The study included a rare six-month follow-up to test for knowledge but did not compare fact boxes to text alone.

Overall, the fact box evaluation literature has not focused on testing the format while controlling for content. The outcomes were diverse, ranging from message liking to comprehension to treatment decisions. So far, 10 papers contain 11 experiments, and nine are convenience samples. Two of the studies registered a study protocol, none pre-specified statistical tests, and all experiments reported better outcomes in the fact box condition. Owing to a lack of text-only control condition with the same content, a lack of controlled trials, and the potential in this area for unpublished studies with no effect, a meta-analysis of this literature might not accurately represent the true effect size of fact boxes on comprehension relative to other formats. The area needs a robust study testing different scenarios that vary in the ratio of benefits to harms, across different presentation formats, keeping the content the same.

## 1.2. Evidence summary formats

Many similar evidence summary formats have been proposed recently, including the: Cochrane summary of findings [23]; evidence summary [24]; plain language summary or significance statements, etc. These summaries share the key feature of being short (less than a page). Otherwise, their format and content vary widely, and it is not clear which features lead to better outcomes. Therefore, the current study focuses on the promising fact box format using tables to represent harms and benefits for different treatment options.

## 1.3. Current study

The current pre-registered study was based on a pilot and offers high power, two medical topics (preventative and therapeutic), a follow-up assessment, two quality checks on the key outcome, and a representative UK sample. The outcome measures were designed to assess the usability and universality of the experimental messages [8]. The key aim was testing whether fact boxes led to better comprehension compared to narrative text with the same information. After about one month, knowledge was tested again, to see if any difference in initial comprehension led to longer-term improvements in recall memory. Additionally, after the recall measure, participants were shown the summary again and completed the comprehension test a final time, to look for practice effects with the tabular format. Another aim was to evaluate treatment decisions based on summary format. Whether improved comprehension led to changes in treatment will depend on the effectiveness and side effects of the described treatment. In this study, both treatments have some benefits but also harms. We suggest it is not objectively good or bad if treatment decision frequencies change between conditions, but it is important to recognize these changes to inform medical practice.

## 1.4. Hypotheses

H1: the fact boxes will lead to greater comprehension than the text-only control at time 1;
H2: the fact boxes will lead to greater knowledge (recall memory) than control at time 2;
H3: the fact boxes will lead to greater comprehension than control at time 2;
H4: the fact boxes will lead to feeling more informed than control;
H5: the fact boxes will not change treatment decisions compared to control;
H6: the fact boxes may lead to more consistent decisions over time than text alone;
H7: the fact boxes will lead to more engagement than the control;

H8: H1–6 will not differ between medical topics (preventative and therapeutic); and

H9: greater comprehension will be shown by high- compared to low-numeracy individuals at time 1.

# 2. Methods

## 2.1. Sampling

A pilot study served to test the procedure and iterate the materials (see below). Participants were recruited online from the survey company YouGov to participate in a two-part survey experiment. Respondents were matched to the UK census by weighting on age, gender, social class, region, level of education, how respondents voted at the previous election, how respondents voted at the European Union referendum, and their level of political interest. Participants were compensated with the equivalent of £5 in vouchers for about 12 min in the study. The first pilot study ($N = 103$) had a survey completion time of $M$ (s.d.) = 767 (376) seconds for time 1; see https://web.archive.org/web/20190421035047/https://yougov.co.uk/about/panel-methodology/.

Exclusion criteria: not a UK resident. No outliers were excluded. Recruitment ended when the completion target was reached.

## 2.2. Procedure and open data, code, materials

The survey flow, questionnaire images and text, cleaning and analysis code (in $r$), variable codebook and data are all available at the Open Science Framework: https://osf.io/n3r5g and the approved protocol at https://osf.io/zwbp9. In the online survey, participants responded to Likert-type questions, saw a health communication, and answered some questions about the health information. The design was 2 (medical topic) × 2 (message type) between-subjects across two time points. All measures and tasks were shown below in order. The key independent variables were medical topic and message type (each randomized), and the key outcome was comprehension.

### 2.2.1. Medical topic manipulation

Participants were randomized to receive information about one of two topics: antibiotics for ear infection or vaccines for influenza. One treatment was preventative (vaccine) and the other is therapeutic (antibiotics).

### 2.2.2. Attitudes

Baseline attitudes were measured towards different treatment topics: influenza (flu) vaccine; teeth cleaning; genetic testing; antibiotics (randomized order). Participants were asked to rate each of three items (presented in a randomized order) on whether they feel it is: effective; safe; based on high-quality science. Each item was measured on Likert scales ranging from 1 (*not at all*) to 5 (*very much*). Mean composites were formed within each treatment. The pattern of attitudes provided a descriptive context for the preventative and therapeutic scenarios selected in this test.

### 2.2.3. Fact box manipulation

Participants were randomized to either a fact box summary or a text-only summary (control), and both conditions contained the same information. Prior to seeing the summary, a brief vignette (less than 210 words in each topic) explained the context of the disease, treatment, and efficacy studies. Then participants saw the summary and continued to the next page. On this second page, they saw the summary again and answered seven comprehension questions. On the third page, they saw the summary again and answered five comprehension questions.

The fact boxes were factually accurate and were built from Cochrane systematic reviews [25]. The influenza vaccine fact box was modified to present central estimates rather than ranges for several values. These two topics and fact boxes were chosen owing to several advantages: (i) they provide coverage of two types of medical treatments: preventative and therapeutic, providing greater generalizability; (ii) the two examples have similar complexity between harms and benefits; (iii) each summary reflects a balance of harms and benefits without a clear best option; (iv) the fact boxes were already summarized from systematic reviews; and (v) these topics would be appropriate for extension in future work to policy-level decisions. The paragraphs beginning 'assume …' were not part of the original

(*a*)   The numbers below are for children 0–15 years of age with an acute middle ear infection who either received antibiotics or placebo (sugar pill) over a period of 7–14 days.

|  | 100 children who took placebo (sugar pill) | 100 children who took antibiotics |
|---|---|---|
| **Benefits** | | |
| How many children had pain 4–7 days after diagnosis? | 11 | 9 |
| How many children continued to have impaired hearing four to six weeks after diagnosis? | 40 | 40 |
| How many children experienced a ruptured (perforated) eardrum as a result of the infection? | 4 | 1 |
| **Harms** | | |
| How many children experienced adverse effects (e.g., vomiting, diarrhoea or rash) during treatment? | 19 | 26 |

(*b*)   The numbers below are for adults aged 60 or older who were observed for one year. Older adults with placebo received an injection with a saline solution (no vaccine) instead of the influenza vaccine.

|  | 1000 older adults with placebo (saline) | 1000 older adults with influenza vaccine |
|---|---|---|
| **Benefits** | | |
| How many older adults developed confirmed influenza (flu)? | 85 | 31 |
| How many older adults developed an influenza-like illness? | 68 | 52 |
| How many older adults died from any cause? | 11 | 9 |
| **Harms** | | |
| How many older adults experienced pain or tenderness in their arm? | 37 | 130 |
| How many older adults experienced redness, swelling, or hardening at the injection site? | 9 | 71 |

**Figure 1.** Fact box messages. (*a*) Middle ear infections in children (Acute Otitis Media). (*b*) Influenza (flu) vaccination for older adults.

fact boxes and serve as task instructions. The text-only control messages were written to match all content of the fact boxes in narrative form for a stringent comparison. All pages with the summary messages were separately timed for participant engagement. No timing exclusions were made (figures 1 and 2).

### 2.2.4. Objective comprehension

The key outcome was objective comprehension of the study results. Of 12 questions, eight were multiple-choice questions with five options, and four were open-response requests for numbers. The questions were designed to cover both gist and verbatim understanding, were adapted from a range of previous papers using comprehension measures, and were pilot tested (see below). Additionally, an objective, validated measure of numeracy (see below) served as a quality check: it is expected to relate moderately positively with the comprehension measure because performance on both requires overlapping skills

| Middle ear infections in children (acute otitis media) | Influenza (flu) vaccination for older adults |
| --- | --- |
| Assume you found the following information about antibiotics for treating inner ear infections in children. Please read the information to consider whether or not you would decide to choose antibiotic treatment for your child | Assume you found the following information about vaccines to prevent influenza in older adults. Please read the information to consider whether or not you would decide to recommend a vaccine for an older relative |
| The numbers below are for children 0–15 years of age with an acute middle ear infection who either received antibiotics or placebo (sugar pill) over a period of 7–14 days | The numbers below are for adults aged 60 or older who were observed for one year. Older adults with placebo received an injection with a saline solution (no vaccine) instead of the influenza vaccine |
| Benefits: of the 100 children who took antibiotics, nine children had pain 4–7 days after diagnosis compared to 11 out of the 100 children who took the placebo (sugar pill). There was no difference between groups in how many children had impaired hearing 4–6 weeks after diagnosis (40 in each group). One out of 100 children who took antibiotics had a ruptured (perforated) eardrum as a result of the infection, compared to 4 out of 100 children who took placebo | Benefits: of 1000 adults ages 60 and older who received the influenza vaccine, 31 developed confirmed influenza (flu) over the next year, compared to 85 out of 1000 older adults who received a placebo. Of older adults receiving the influenza vaccine, 52 out of 1000 developed an influenza-like illness, while 68 out of 1000 did in the placebo group. In the influenza vaccine group, 9 out of 1000 died of all causes over the next year, while 11 out of 1000 died in the placebo group |
| Harms: 26 out of 100 children taking antibiotics experienced an adverse effect (e.g., vomiting, diarrhoea or rash), compared to 19 out of 100 children who took the placebo | Harms: of the 1000 older adults receiving the influenza vaccine, 132 experienced pain or tenderness in their arm, compared to 37 out of 1000 in the placebo group. In the influenza vaccine group, 71 out of 1000 experienced redness, swelling, or hardening at the injection site, compared to 9 out of 1000 in the placebo group |

**Figure 2.** Text-only messages (control).

such as sustained attention and facility with numbers. The comprehension questions are highly similar across the two medical topics and differ only in what number or comparison is specified, because each topic contains a different intervention and also outcomes. Great care was taken to standardize the questions between conditions. Each of the 12 items was scored exactly correct or not and a composite of comprehension was calculated from the proportion of correct items. A representative question is: 'out of 100 children with a middle ear infection who took antibiotics, how many experienced a ruptured eardrum?' (open response: correct = '1').

### 2.2.5. Decision

Participants were prompted to imagine that a relative is making a decision on this subject and the participant was asked whether they would recommend the treatment. The response options were phrased around the particular intervention, e.g. antibiotics or vaccine, and are only summarized here: yes; no; unsure; no difference.

### 2.2.6. Informed decision-making

Participants reported whether they felt they understood the available options; the benefits; and the risks and side effects, rated from 1 (strongly disagree) to 7 (strongly agree). These items constitute the informed subscale of the Decisional Conflict Scale [26] and were averaged into a composite.

### 2.2.7. Message engagement

Participants rated how much others would want to read the message; whether the participant is interested in the information; and whether they like how it is presented, rated from 1 (not at all) to 5 (very much), adapted from [27]. Participants also reported how reliable the information is, 1 (not at all) to 5 (very reliable), how trustworthy it is 1 (not at all) to 5 (very trustworthy), and optionally wrote a comment in an open-response field.

### 2.2.8. Objective numeracy

Participants completed the adaptive Berlin Numeracy Test [28] to measure facility with numbers and mathematical operations. This served as a quality check for the comprehension measure: numeracy and comprehension were expected to relate moderately positively (see Objective comprehension). Of four numeracy questions, participants completed either two or three depending on their accuracy, and then individuals were sorted into numeracy categories between 1 and 4. Each correct answer was a free-response number. Only exact values were scored correct.

### 2.2.9. Demographics

Participants reported their age, gender, ethnicity/race, highest completed education, perceived social status by clicking one of 10 rungs on a ladder [29], and general health rated from 1 (poor) to 4 (excellent). These measures were used to describe the generalizability of the results.

### 2.2.10. Time 2

After five weeks, participants were emailed with the invitation to the second survey. They completed the decision item again. To measure knowledge (recall memory), they then saw instructions and completed the objective comprehension test (12 items) without any summary information (no message). Then to measure comprehension, they completed the 12 items again in the same format as in time 1 (seeing the evidence summary on each page).

## 2.3. Analysis plan

### 2.3.1. Comprehension

The primary analysis tested whether objective comprehension differs between the fact box and the text control across both topics using ANOVA at time 1 and ANCOVA at time 2 (using time 1 comprehension as a covariate). Using this design, no *post hoc* tests were required because the main effects revealed any differences between means. The choice of AN(C)OVA focused the tests and results on the differences between conditions (as opposed to the change over time) and allowed for easier interpretation of power analyses and results. The fact box was expected to lead to higher comprehension than the text-only control, so this hypothesis was evaluated with a one-tailed test. No distribution shape requirements were used for the main test, as the distribution of comprehension was adequately normal in the pilot (skew = −1.6, kurtosis = 2.3; non-normal defined here as greater than 3). The main study items were also improved based on the pilot (see below), so the distribution was expected to become more normal.

### 2.3.2. Design

Alpha was set initially at 0.05. To correct for multiple tests and reduce the chance of false positives, this threshold was reduced to $\alpha = 0.01$. Because the hypotheses are directional and pre-registered, they can be evaluated with one-tailed tests. The study design is 2 (topic) × 2 (message format) between-subjects. As justified and computed below, all hypotheses were powered to at least 0.9 at $\alpha = 0.02$ for one-tailed tests to detect $d = 0.2$ accounting for multiple comparisons and attrition.

## 2.4. Tests

H1: two-way ANOVA of topic and message on time 1 comprehension;
H2: two-way ANCOVA of topic and message on time 2 recall memory with time 1 comprehension as a covariate;

H3: two-way ANCOVA of topic and message on time 2 comprehension with time 1 comprehension as a covariate;

H4: two-way ANOVA of topic and message on time 1 informed subscale;

H5: equivalence test of topic and message on time 1 decision (yes/no);

H6: *t*-test by condition on difference score between time 1 and time 2 decisions (yes/no);

H7: two-way ANOVA of topic and message on engagement;

H8: equivalence testing between medical topics for H1–6; and

H9: one-way ANOVA of numeracy on time 1 comprehension.

### 2.4.1. Effect sizes

Cohen's $d = 0.2$ is a plausible lower bound for the smallest effect sizes of interest for our key variables based on the previous fact box literature and the broader context of psychological effects [30], and additionally is a conservative lower bound for the effect in the pilot study. The published studies on fact boxes consist of 15 studies in 11 papers, including 10 convenience samples. Two of the studies had brief, registered study protocols, none pre-registered their statistical tests, one had a follow-up test (there was no effect of fact boxes at six months compared to control), and all experiments reported better outcomes in the fact box condition across a wide range of control conditions. Only five studies compared comprehension of fact boxes to other non-tabular texts, but only two of those reported $M$'s and s.d.'s. Overall, some of the reported effects were very large ($d > 1$), but these may be inflated owing to a file drawer of unpublished studies and other researcher degrees of freedom. The pre-registered pilot study was underpowered to estimate effect sizes, but ANOVA suggested $d = 0.45$ in the predicted direction (fact boxes led to more comprehension than text alone).

### 2.4.2. Power analysis and sample size

For H1, H4 and H9 (all have four groups), G*Power [31] estimates that $n = 1720$ are needed in ANOVA to detect an effect of $F = 0.1$ (Cohen's $d = 0.2$) at 0.9 power and $\alpha = 0.02$ (one-tailed test). For H2 and H3, G*Power estimates that $n = 1305$ are needed for a $2 \times 2$ ANCOVA to detect an effect of $F = 0.1$ (Cohen's $d = 0.2$) at 0.9 power and $\alpha = 0.02$ (one-tailed test). H5 and H8 were conducted with equivalence tests using the *toster* package in R [32]. H8 is based on two-tailed tests between medical topics for H1–6. A power analysis with *toster* using $\alpha = 0.02$, power $= 0.9$ and upper/lower bounds for Cohen's $d = \pm 0.2$ yielded $n = 342$. All tests were performed in R (R Project for Statistical Computing, RRID:SCR_001905).

The ANOVA estimate was largest and was used for a minimum sample size. Longitudinal attrition is estimated at 25% from online surveys in this population with similar tasks, length and compensation, so 34% more participants were recruited, ($n * 1.34) * 0.75 \sim n$, final $N = 2305$ (576 per $2 \times 2$ cell).

### 2.5. Pilot study

A pre-registered pilot study ($N = 103$) on Prolific tested the proposed methods, estimated the effect size of H1, determined appropriate payment, and provided a quality check, e.g. of distribution normality: pre-registration, data, code and materials at https://osf.io/n3r5g. Seven participants did not complete the comprehension measure and were excluded (6.3%; original $N = 110$).

Participants were randomized to topic and message type and completed the proposed measures. All measures and procedures functioned as planned. There was a slight ceiling effect for comprehension, $M$ (s.d.) $= 5.89$ (2.06), range 0–8, skew $= -1.6$, kurtosis $= 2.3$; non-normal defined as skew or kurtosis greater than 3. An item response theory analysis served to estimate the difficulty and quality of each item. To improve the measurement of the latent construct, and increase difficulty and therefore normality by reducing the ceiling effect, three changes were made before the main study: (i) four more questions were added, designed after the highest quality questions from the item response theory analysis; (ii) the multiple choice questions were changed from four to five response options; and (iii) four multiple choice questions were changed to open-response (exact number required). The pre-registered pilot study was underpowered to estimate effect sizes, but a one-way ANOVA of message type suggested $F_{3,101} = 5.21$, Cohen's $d = 0.45$: as predicted, fact boxes led to more comprehension than text alone.

## 2.6. Deviations from stage 1 registered report

### 2.6.1. One pilot study

Two pilots were proposed in stage 1 but only one was necessary because the distributions of item comprehension were sufficiently normal.

### 2.6.2. Follow-up interval

The approved delay between timepoints was 20 days, but there were unanticipated delays in implementing the survey instrument and slow follow-up enrolment, final interval $M$ (s.d.) = 44.6 (6.8) days.

### 2.6.3. Scientific source omission

The original design included stating the source of the evidence to participants in all conditions (see source text below). However, this text was accidentally dropped by the survey company and the omission was not noted by the authors. See the spontaneous comments about source in the Results and Discussion. Omitted text by condition:

 (i) [ear] sources: Institute for Quality and Efficiency in Health Care, May 2013; Lieberthal *et al.* (2012), doi: 10.1542/ peds.2012–3488; Venekamp *et al.* (2013), Cochrane Database Syst. Rev., 1, CD000219;

 (ii) [flu] sources: RKI guide: Influenza (Pt. 1). 2016; Buda *et al.* Epidemiological report on influenza in Germany 2014/2015; Jefferson *et al.* Cochrane Database Syst. Rev. 2010;2:CD004876; Darvishian *et al.* J. Clin. Epidemiol. 2014;67(7):734–44; Beyer *et al.* Vaccine 2013;31(50):6030–3. Updated: 2016.

### 2.6.4. Analyses

H5 implied a two-way equivalence test, but equivalence tests are one-way (Lakens [32]). Two separate one-way equivalence tests were performed.

# 3. Results

Fact boxes led to greater comprehension than text-only controls, $M$ (s.d.) = 79.6% (23.1) versus 69.7% (27.6) correct, respectively. H1: a two-way ANOVA with format and topic predicting comprehension showed that fact boxes were better understood, $F_{2,2302} = 113$, $p < 0.0001$ (figure 3 and table 1). Ear messages were better understood, $F_{1,2302} = 62$, $p < 0.0001$. The effect size of format on comprehension was Cohen's $d = 0.39$ (95% confidence interval (CI): 0.31–0.47). Raw means, standard deviations and zero-order correlations between key variables are shown in the electronic supplementary material, table S1.

## 3.1. Decision for treatment

Participants indicated whether they would personally support the medical intervention for a member of their family. The decision for treatment as recoded into 1 (yes) and 0 (all other responses) for the main analyses (table 1). H5 was examined with two equivalence tests [32] using alpha = 0.01 and upper and lower bounds of 0.2 for the smallest effect size of interest. There was no difference between fact boxes and text-only controls, $M$ (s.d.) = 63.4% (48.2) versus 61.3% (48.7), respectively, Welch's $t_{2296} = 1.05$, $p = 0.29$. Participants in the flu conditions chose treatment more often than those in the ear conditions, $M$ (s.d.) = 80.6% (39.5) versus 44.8% (49.7), respectively, Welch's $t_{2225} = 19.2$, $p < 0.001$. Electronic supplementary material, table S2 and figure S4 show all decision options (e.g. unsure) by format and topic. There was no difference in how many people selected 'unsure' between formats (see Exploratory analyses).

## 3.2. Trust

The trust measure was a composite of two items, Pearson's $r_{2303} = 0.87$. Participants reading both the fact boxes and the text-only controls reported moderate trust in the evidence, $M$ (s.d.) = 3.56 (0.85) versus 3.59 (0.85) out of 5, respectively. A two-way ANOVA of format and topic showed no main effect of format, $p = 0.47$. However, participants in the flu conditions reported more trust than the ear conditions, $M$ (s.d.) = 3.67 (0.84) versus 3.48 (0.84), respectively, $F_{1,2302} = 31.2$, $p < 0.0001$.

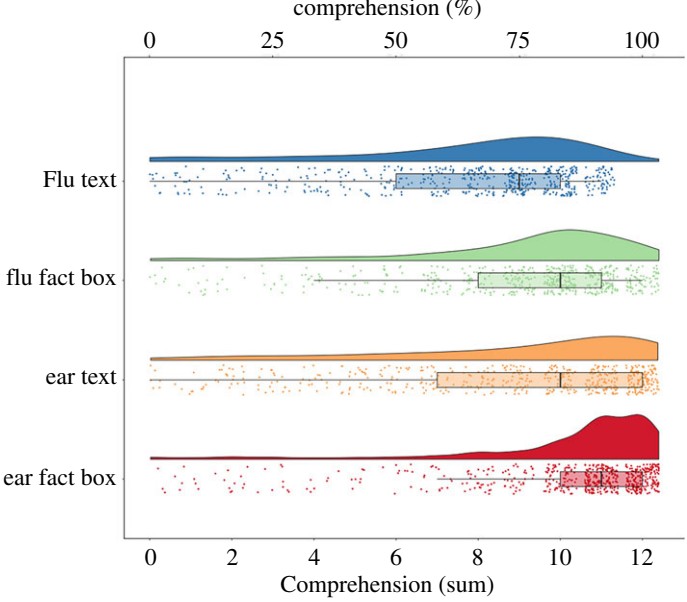

**Figure 3.** Baseline comprehension by experimental condition shown as raincloud plots [33]. The boxes indicate the interquartile range (25–75%), the whiskers the values within 1.5 times that range, and the vertical black lines the medians.

**Table 1.** Effects of condition on baseline comprehension, treatment decision and message evaluation. (Note: treatment decision is coded here as 1 (yes) and 0 (all other responses).)

| M (s.d.) | fact box | text | ear fact box | ear text | flu fact box | flu text |
|---|---|---|---|---|---|---|
| comprehension (%) | 79.6 (23.1) | 69.7 (27.6) | 83.7 (22.7) | 73.7 (27.5) | 75.5 (22.8) | 65.4 (27.0) |
| treatment decision (%) | 63.4 (48.2) | 61.3 (48.7) | 46.9 (49.9) | 42.5 (49.5) | 80.4 (39.7) | 80.8 (39.4) |
| informed (1–7) | 5.51 (1.15) | 5.43 (1.17) | 5.39 (1.13) | 5.34 (1.11) | 5.64 (1.16) | 5.53 (1.22) |
| engaged (1–5) | 3.70 (0.92) | 3.45 (1.00) | 3.65 (0.93) | 3.45 (0.97) | 3.75 (0.92) | 3.45 (1.03) |
| trust (1–5) | 3.56 (0.85) | 3.69 (0.85) | 3.43 (0.85) | 3.53 (0.83) | 3.70 (0.82) | 3.65 (0.87) |
| n | 1177 | 1128 | 599 | 576 | 578 | 552 |

## 3.3. Informed

Participants in both the fact box and text-only conditions felt informed about the content, 3-item composite $M$ (s.d.) = 5.51 (1.15) versus 5.43 (1.17) out of 7, respectively. H4 specified a two-way ANOVA of format and topic, and it showed no main effect of format, $F_{2,2302} = 2.88$, $p = 0.09$. However, participants in the flu conditions reported feeling more informed than the ear conditions, $M$ (s.d.) = 5.59 (1.19) versus 5.36 (1.12), respectively, $F_{2,2302} = 21.7$, $p < 0.0001$.

## 3.4. Engaging

Fact boxes were rated more engaging than the text-only controls, $M$ (s.d.) = 3.70 (0.92) versus 3.45 (1.00) out of 5, respectively. H7 specified a two-way ANOVA with format and topic predicting engagement. Fact boxes were more engaging, $F_{2, 2302} = 36.8$, $p < 0.0001$. There was no effect of topic, $p = 0.21$.

## 3.5. Objective numeracy

The adaptive Berlin Numeracy Test sorted participants into four categories. $n$s per numeracy group: 1 (841), 2 (657), 3 (289), 4 (518). To satisfy H9, we ran an ANOVA predicting baseline comprehension from the four numeracy categories, and higher numeracy was associated with greater comprehension, $F_{3,2303} = 279$, $p < 0.0001$ (figure 4).

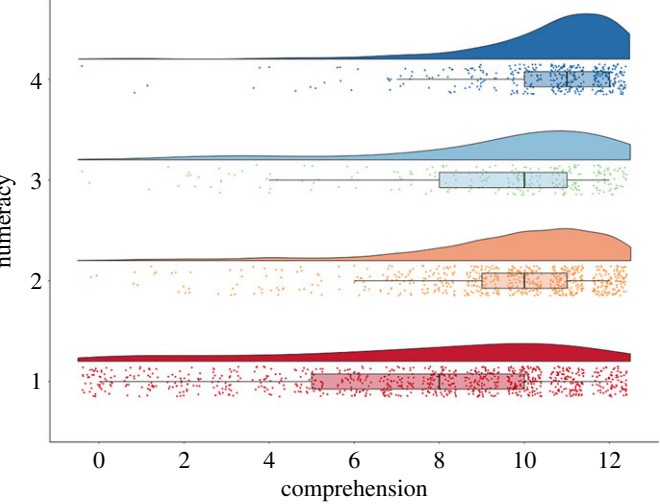

**Figure 4.** Objective numeracy and objective comprehension at baseline shown as raincloud plots [33]. The boxes indicate the interquartile range of that row (25–75%), the whiskers the values within 1.5 times that range, and the vertical black lines the medians.

**Table 2.** Effects of condition on follow-up recall, treatment decision and message evaluation. (Note: T2, time 2 (follow-up). Treatment decision is coded here as 1 (yes) and 0 (all other responses).)

| M (s.d.) | fact box | text | ear fact box | ear text | flu fact box | flu text |
|---|---|---|---|---|---|---|
| recall T2 (%) | 29.8 (17.1) | 27.7 (17.5) | 27.6 (16.0) | 25.6 (16.3) | 32.1 (17.9) | 30.0 (18.5) |
| comprehension T2 (%) | 77.5 (26.1) | 65.4 (30.7) | 81.6 (25.5) | 69.0 (31.1) | 73.2 (26.1) | 61.5 (29.8) |
| treatment decision T2 (%) | 65.4 (47.6) | 58.5 (49.3) | 47.0 (49.9) | 36.6 (48.2) | 85.1 (35.6) | 82.7 (37.8) |
| engaged T2 (1–5) | 3.45 (1.00) | 3.19 (1.03) | 3.32 (1.00) | 3.18 (1.01) | 3.57 (.98) | 3.20 (1.05) |
| trust T2 (1–5) | 3.49 (0.89) | 3.46 (0.93) | 3.34 (0.89) | 3.41 (0.89) | 3.65 (0.86) | 3.51 (0.97) |
| n | 861 | 805 | 445 | 423 | 416 | 382 |

See the electronic supplementary material for additional raincloud plots, e.g. of trust, feeling informed and engagement by condition.

## 3.6. Follow-up time point

The participants were invited to a follow-up study after five weeks. Of 2305 baseline participants, 1666 completed the follow-up (retention = 72.3%). The confirmatory tests were pre-registered for 0.9 power and 75% retention, so the follow-up analyses ended up powered slightly less than 0.9 for main effects of Cohen's $d \leq 0.2$. In the follow-up, participants made another treatment decision and completed a recall test of the comprehension questions, both without seeing the evidence again. The recall items including open response were strictly scored as exactly correct or not, like at baseline. The participants then saw the same evidence as at baseline and completed the comprehension questions for a third time. These measures allow the testing of decision consistency over time by condition, recall without materials, and comprehension again with the same materials.

Fact boxes led to more recall after six weeks than text-only controls, M (s.d.) = 29.8% (17.1) versus 27.7% (17.5) correct, respectively. H2: a two-way ANCOVA predicting recall from format and topic with time 1 comprehension as a covariate showed that fact boxes were better recalled: $F_{1,1662} = 7.62$, $p = 0.006$. Higher comprehension at time 1 predicted higher recall, $F_{1,1662} = 450$, $p < 0.0001$, and there was no interaction, $p = 0.53$ (table 2 and figure 5). The effect size of format on recall was Cohen's $d = 0.12$ (95% CI: 0.09–0.15).

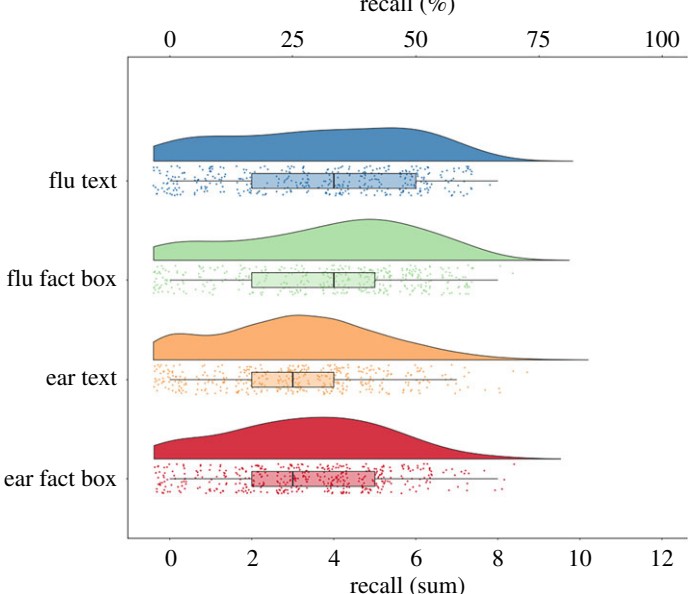

**Figure 5.** Recall by experimental condition shown as raincloud plots [33]. The boxes indicate the interquartile range (25–75%), the whiskers the values within 1.5 times that range, and the vertical black lines the medians.

Similarly, after participants saw the evidence again, fact boxes led to greater follow-up comprehension than text alone, $M$ (s.d.) = 77.5% (26.1) versus 65.4% (30.7), respectively (figure 5). H3: a two-way ANCOVA predicting follow-up comprehension from format and topic with time 1 comprehension as a covariate showed that fact boxes were better understood: $F_{1,1662} = 57$, $p < 0.0001$. Higher comprehension at time 1 predicted more comprehension at time 2, $F_{1,1662} = 1814$, $p < 0.0001$, and there was no interaction, $p = 0.74$ (table 2; electronic supplementary material, figure S6). The effect size of format on follow-up comprehension was $d = 0.43$ (95% CI: 0.33–0.52), similar to the estimates from time 1 and the pilot ($d = 0.39$ and 0.45, respectively).

The last confirmatory test examined the change in treatment decisions over time (H6). Treatment decisions were coded as 1 (yes) and 0 (all other responses). Most decisions (76.7%) did not change between timepoints. T1 was subtracted from T2 (tables 1 and 2). H6: a $t$-test by topic and format on the difference score found no effect of format, $p = 0.12$, but did reveal an effect of topic, $F_{1,1663} = 6.87$, $p = 0.009$. Decisions for treatment decreased 3.34% in the ear conditions and increased 2.88% in the flu conditions.

## 3.7. Exploratory analyses

### 3.7.1. Recall by item type

Recall of the free-response items (1, 8, 9 and 10) was at floor in both conditions, fact box $M$ (s.d.) = 0.10 (0.30), text $M$ (s.d.) = 0.12 (0.35) (hypothetical range 0–4). See the electronic supplementary material, figure S7 for the raw histograms. This floor effect was not surprising given the six-week delay and only scoring exact answers as correct. Therefore, the eight multiple-choice items are probably driving the differences by condition on recall, multiple choice items fact box $M$ (s.d.) = 3.48 (2.03), text $M$ (s.d.) = 3.20 (2.05).

### 3.7.2. Education and numeracy

The sample was not large enough to ensure high power for moderation effects, particularly for interactions that are attenuating rather than cross-over, and sufficient statistical power for testing moderation is an area of unresolved debate [34]. Therefore, we focus on descriptive results in this section. It was important to assess whether the beneficial effect of fact boxes was consistent across participant education (figure 6). Highest attained education was recorded on a 20-option scale and divided into four categories because they are of comparable size. The UK categories and $n$ are provided with the US equivalent in parentheses: primary (elementary or less; $n = 349$); GCSE (partial high school or less, $n = 648$); BA (bachelors' or less, $n = 716$); MA+ (masters' or higher, $n = 592$). Visual inspection suggests that there was a comparable benefit of fact boxes over text at all levels of education.

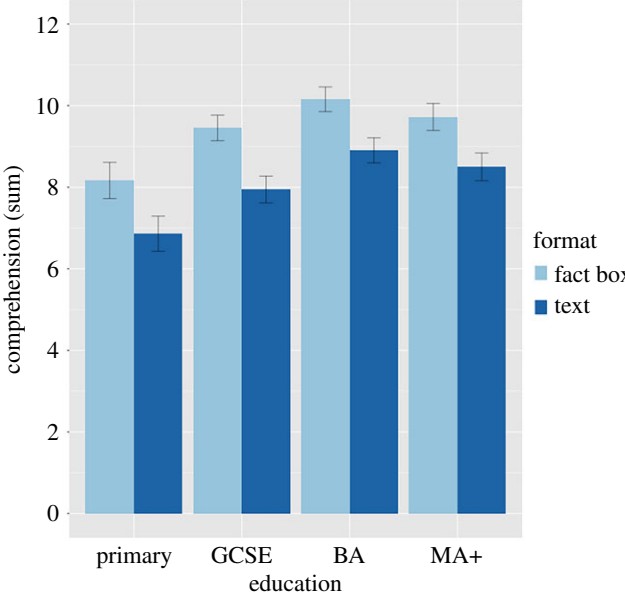

**Figure 6.** Education, format and objective comprehension at baseline. Error bars are 95% CIs based on standard errors.

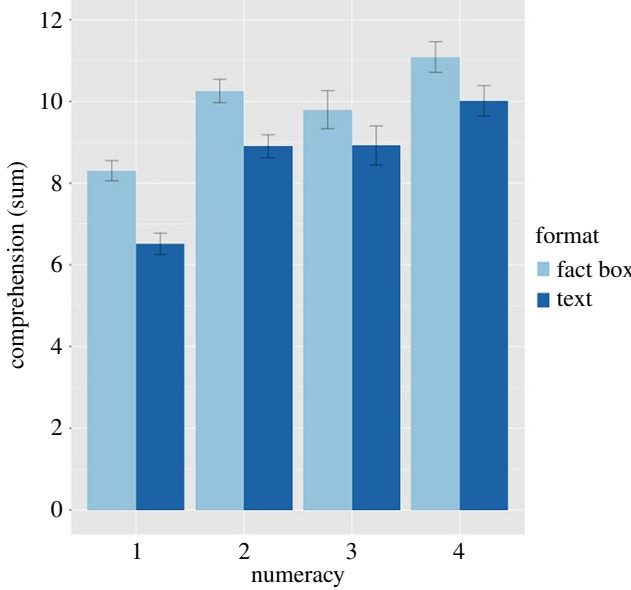

**Figure 7.** Numeracy, format and objective comprehension at baseline. Error bars are 95% CIs based on standard errors.

Fact boxes also appeared to improve comprehension across the levels of numeracy, and perhaps most for the lowest numeracy participants (figure 7). Numeracy and the four-group education composite were only correlated Spearman's rho $r_{s2303} = 0.16$, $p < 0.0001$ (exploratory), so the numeracy-comprehension pattern in figure 7 is probably distinct from the education result in figure 6.

### 3.7.3. Decision

There was no difference in choosing the option 'unsure' for treatment decision between message formats, equivalence test Welch's $t_{2294} = -0.36$, $p = 0.72$ (exploratory).

### 3.7.4. Attitudes about medical treatments

Attitudes about antibiotics, vaccines, genetic testing and tooth brushing were moderately positive (all composite means between 3.35 and 3.99 out of 5; see the electronic supplementary material, table

S1). In the flu condition, positive attitudes about vaccines were associated with deciding for treatment, $r_{pb}(1128) = 0.36$, $p < 0.0001$ and with comprehension, $r_{pb}(1128) = 0.22$, $p < 0.0001$ (both point biserial correlations were exploratory). Similarly, in the ear condition, positive attitudes about antibiotics were associated with deciding for treatment, $r_{pb}(1173) = 0.13$, $p < 0.0001$, and with comprehension, $r_{pb}(1173) = 0.26$, $p < 0.0001$ (both point biserial correlations were exploratory).

### 3.7.5. Attrition

Participation in the follow-up survey appeared slightly higher for participants who saw a fact box (73.2%) rather than text only (71.4%).

## 3.8. Open-response comments (baseline)

There were two optional comment fields at baseline, one midway through the survey and one right before the end. Both asked for general, open-ended feedback about the survey or evidence. The lead author read the responses and developed thematic categories. The themes were then iterated by both authors, yielding 14. Each entry was recorded by both authors separately, and agreement was adequate: 2818 responses from 1519 participants, kappa = 0.77, agreement = 83.9%. All disagreements were resolved through discussion. Table 3 shows the frequency of each category by format condition and in total. There were no large differences between formats. The key themes are explored below.

### 3.8.1. Misinterpretation of the 'out of 100 people' format

About 7.9% wrote about sample size, frequently saying that 100 or 1000 (depending on topic) were too few study participants to make confident claims. The participants seemed to misunderstand that 'out of 100/1000' was an expected frequency, not a report of the sample size used to evaluate the interventions. The underlying meta-analyses included more individuals.

### 3.8.2. Recognition of limited benefit

About 7.3% wrote about the balance of harms and benefits, and many wrote that they were surprised how modest the treatment benefits were, particularly about the ear infection condition. For example, a 65-year-old man wrote: 'Lots of people take antibiotics when there is no need to. If info was given like this it would make it easier for people to decide whether or not antibiotics are really needed'.

### 3.8.3. No source provided for the evidence

About 3.2% talked about the lack of a source for the evidence, for example, saying they could not rate the trustworthiness of the information because of it. The omission of the source was a technical error, but it had the side-effect of revealing that individuals spontaneously mentioned the importance of the source when it was not provided.

### 3.8.4. Stress and threat

To the surprise of the authors, 15.0% left comments about their performance on the comprehension task and objective numeracy questions, frequently seeming distressed or trying to justify a low score. Comments referred to their mathematical education, age, medical status or cognitive abilities. Some participants also spontaneously wrote that they understood the material, but that it would be too difficult for others. An informal reading of the comments in the follow-up survey identified the same themes as in the baseline.

# 4. Discussion

Medical information provided in fact box format was understood better than the same content provided in text across two topics, two timepoints, and across diverse levels of education and numeracy. The effects of format on comprehension ($d_{T1} = 0.39$, $d_{T2} = 0.43$) are large for this kind of format intervention, particularly as the content was very consistent between text and table formats. Fact boxes also led to slightly better recall (knowledge) after six weeks ($d = 0.12$; 29.8% versus 27.7% for text alone). Higher scores are better,

**Table 3.** Open responses coded into themes ($n = 2818$). (Note: total % is based on the 1519 participants who left at least one comment. Respondents with two comments were counted twice. Indented rows show specific categories that are also included in the non-indented subtotals.)

| themes | fact box $n$ | text $n$ | total % | explanation |
|---|---|---|---|---|
| Format | | | | |
| positive | 22 | 30 | 3.4 | evidence was easy to read |
| negative | 116 | 104 | 14.5 | evidence was hard to read |
| requested graphics | 37 | 33 | 4.6 | suggested using graphics |
| Evidence | | | | |
| positive/neutral | 50 | 61 | 7.3 | summarizing or considering the evidence |
| negative | 159 | 159 | 20.1 | |
| skeptical | 9 | 8 | 1.1 | skepticism about the evidence |
| wanted statistical significance | 7 | 7 | 0.9 | mentioned differences could be owing to chance |
| wanted source for data | 29 | 19 | 3.2 | mentioned no source was given for the evidence |
| wanted more info on interventions or outcomes | 35 | 41 | 5.0 | details unclear for outcomes (e.g. severity) or treatment (e.g. dosage) |
| wanted more info on participant subgroups | 19 | 10 | 1.9 | previous medical history, outcomes between age groups/sexes, etc. |
| wanted bigger sample size | 56 | 64 | 7.9 | believed 'out of 100', 'out of 1000' was sample size |
| wanted population effects | 4 | 10 | 1.0 | antibiotic resistance or herd immunity |
| Other | | | | |
| difficulty, maths | 179 | 167 | 15.0 | difficulty or justification, mostly about numeracy test |
| anecdotes, uncoded | 133 | 123 | 11.1 | anecdotes and uncoded text |
| none | 633 | 652 | 55.7 | none, thank you, similar or gibberish |

but it is unknown whether this size effect would lead to meaningful differences in informed decision-making. There was a floor effect on recall, probably indicating high difficulty. Exploratory analyses suggested that the effect of format on recall was not just owing to forgetting specific numbers.

The comprehension items were designed to be ecologically valid: to reflect the key information that a person would need to make an informed decision between options. It would be possible to craft more difficult items, for example by introducing deliberate wording distractions or asking individuals to perform complex arithmetic, but this study focused on realistic questions. Because of the potential ceiling effect on the comprehension measures, these effect sizes potentially underestimate the benefit of fact boxes over text for more complex materials.

There is a particular need to identify summary formats and decision aids that support low-numeracy and low-education individuals [35], so it is heartening to see the consistency of the benefit across education levels and possibly an increased benefit for the lowest-numeracy individuals. There is no evidence here of widening existing inequalities, which is a concern around increasing information-provision and shared decision-making in medicine [36].

As expected, higher numeracy individuals understood the materials better. There are probably two effects driving this pattern. First, differences in underlying abilities to process information and numbers [35] probably increased both numeracy and comprehension. Second, differences in task motivation and engagement could create this pattern separate from differences in ability. Participants vary in their interest and commitment to difficult survey tasks, and those participants who were less motivated or engaged may have performed poorly on both the comprehension and numeracy questions, independent of ability. Future research using the Berlin Numeracy Test might benefit from including the three-item

Schwartz scale to limit the risk of numeracy floor effects in the general population [28,37]. Ability and motivation could also separately explain other positive associations, such as between educational attainment and comprehension, and between positive attitudes about each treatment and comprehension in both medical conditions (exploratory). For this last finding, participants may have remembered information better when it was consistent with their prior beliefs (motivated reasoning).

## 4.1. Treatment decisions, trust, feeling informed and engagement

Decision aids that boost comprehension might also affect user attitudes or intentions. In contexts like medical decisions where there is no correct answer about treatment, these changes might be an unwelcome side effect of a new format. Here, the fact box format did not affect trust, feeling informed, or decisions for treatment relative to text-only controls. Fact boxes were rated as more engaging, which may help individuals pay attention to them. Feeling informed was higher for the flu than ear infection conditions, possibly because the relative balance of benefits versus harms was less ambiguous in the flu messages (more positive), and/or because trust was higher for flu than ear infection messages.

To speculate, the format may not have changed treatment intentions for a few reasons, including the ambiguity of both scenarios (the treatments had both harms and benefits), pre-existing beliefs and attitudes, or that overall comprehension was sufficient to make decisions in either format. The flu vaccine had a better benefit-to-harm ratio compared to the antibiotics, and there were nearly twice as many decisions for treatment in the flu compared to ear infection conditions. See the electronic supplementary material, table S1 to see means, standard deviations, and zero-order correlations between treatment decision and other key variables: all associations with treatment decision were small and positive.

## 4.2. Generalizability and validity

The study extends prior work on fact boxes by including two medical topics to evaluate consistency across different topics, pre-registered analyses, and an equivalent control condition to isolate the effect of format (table versus text). It also adds to relatively few studies with longitudinal follow-up or true evidence summaries. The findings are immediately generalizable to the population and context of the current study: UK residents reading about the quantitative harms and benefits of medical choices for family members with a balance of harms and benefits [38]. The results are not a direct test of non-UK populations, non-medical topics, evidence without numbers, decisions about the self-compared to family members, or domains with a different balance of harms and benefits.

The treatment choice was hypothetical in the current study, and performance and preferences might shift in a family member who must make a consequential medical decision. A secondary limitation is validity, based on the novel comprehension questions and wording of the text-only controls. The use of clear, brief text with exactly the same content as the fact boxes was deliberate and constituted a stringent test for fact boxes. Incidentally, the sentences read very similarly to existing evidence summaries [39]. A different choice of text wording may have led to changes in comprehension relative to the current results. For example, a text summary that mixes the order of harms and benefits, or has other complex content, will probably be harder to navigate and would lead to lower comprehension.

## 4.3. Lessons from open response

Several of the themes from the open response comments were revealing and inform the design of future communication formats. First, participants wanted to know more about the source and quality of the evidence. The accidental omission of the source statement may explain why trust in the materials was only moderately positive. After reflection, even had we included the citation to a scientific review, the reference alone might not be sufficient. Communicators should consider explaining the source and quality of the scientific evidence in sentences, as the Harding Center for Risk Literacy does for their publicly available fact boxes [25]. Future evaluations could test the effect of these presentations on trust. Another option is using icons to graphically express the quality of evidence. A recent study examined which quality of evidence icons were best understood by an expert sample of policy makers and practitioners [40]. Expert and general public users had similar goals and comprehension of icons, and both groups widely misunderstood common icons, e.g. those intended to communicate intervention effectiveness or quality of evidence.

Second, some participants misunderstood the source and quality of the represented data, and mistakenly thought that only 100 (or 1000 in the flu condition) individuals took part in the medical research being presented. Keeping the denominator the same while comparing groups is critical for

communications to be understood [13], and using familiar, base-10 denominators is likely to help the reader, e.g. eschewing '312 out of 473' for the equivalent '66 out of 100'. However, while individuals may understand the numbers better with a common denominator, there may be risks to trust and understanding of the quality of evidence if the phrasing does not make clear the underlying sample size. Additionally, researchers may forget (as we did) that most people cannot determine evidence quality from a scientific reference without any explanation. We recommend that both the 'out of X' format and the source of the material be explained more clearly to participants. For example, the Harding Center for Risk Literacy explains the sample size behind the evidence in surrounding text [25] and this is recommended in current evidence-based medicine practices for patient information materials [41]. Future research could test how these summaries affect trust and comprehension of study quality. People may need context as well as numbers.

Third, participants often asked for more information than the data provided, such as the effects on different subgroups (gender, age) or outcome severity, and a few mentioned wanting to know about the population-level effects of individual choices (here: antibiotic resistance and herd immunity from vaccination). These comments are consistent with previous findings that participants regularly want more detail on presented evidence, but it remains unclear how to balance clarity and brevity with comprehensiveness [40]. Broadly, the current insights reinforce the importance of user-centred design during the development of communication materials, in order to find out what outcomes, subgroups and details are desired by the target audience.

Fourth, some participants spontaneously suggested that the use of graphics might enhance the comprehension of the evidence (there was no difference in this suggestion between formats). Some medical fact boxes already use graphics [14]; that study found that graphics were equivalent in comprehension to other formats (e.g. tables versus icon arrays). Some readers will prefer and better understand certain formats. It may be ideal to present multiple formats that the reader could choose between.

Finally, participants left many spontaneous comments indicating stress around the comprehension and numeracy tasks. People differ in their motivation and ability to perform mathematical operations [35]. We assessed numeracy with a difficult, objective measure. One alternative is a subjective measure of numeracy; in general, they capture much of the same variance [42] and may be less stressful. The current results are consistent with the previous literature that more numerate participants extract more accurate information out of risk displays and this numeracy effect is usually not moderated by format, e.g. [43].

# 5. Conclusion

Identifying effective summary formats is fundamental to evidence communication in a wide range of fields. Informed decisions are only possible when individuals understand the potential harms and benefits of different choices. The study offered a high-powered study using a representative UK sample with pre-registered analysis, included two medical topics (preventative and therapeutic) using real data, incorporated a follow-up assessment, and two quality checks on the key outcome (results from a pilot study and the positive association between comprehension and objective numeracy). Fact boxes were understood better and rated more engaging than the text-only controls, and there were no differences for trust or decisions for treatment. We strongly recommend using a fact box format over text alone in similar messages for patients and other populations making individual medical decisions. We also recommend further work on whether incorporating simple graphics may enhance the fact box format (although see [14]).

This study constitutes the best evidence to date for the usefulness of tables for summarizing potential harms and benefits to inform decisions in a medical context. The same format is likely to be of benefit in summarizing findings in scientific papers, particularly in meta-analyses and reviews, such as those used in Cochrane Group summaries of findings [44], economic reports and official statistical releases.

Future work can test the extension of these findings to other populations, other domains, and other decision needs. For example, summarizing policy options is more challenging than individual options owing to complexities such as different effects on different groups [16]. We plan to follow this work with a study adapting these fact boxes for policy options. The two current topics (antibiotic use and vaccination) were chosen as they are appropriate for a national-level guideline decision. For individual health information, we suggest that the evidence now strongly recommends the use of tables over plain text for communicating numerical harms and benefits.

**Research ethics.** The study ethics were approved by the Institutional Review Board of the University of Cambridge, PRE.2018.102.

Data accessibility. The survey flow, questionnaire images and text, cleaning and analysis code in R, codebook, and raw data are all available at the Open Science Framework: https://osf.io/n3r5g/.

Authors' contributions. C.B., M.M. and A.F. designed the studies and materials. C.B. collected the data, wrote the cleaning, analysis, and figure code, and drafted both manuscript stages. M.M. provided a markdown script, training and code review. M.M. and A.F. contributed editing and guidance. All authors gave final approval for publication.

Competing interests. The authors declare no competing interests.

Funding. Funding was provided by the David and Claudia Harding Foundation and the Winton Charitable Foundation.

Acknowledgements. We thank the contributors to R, R Studio and the R packages assertr, here, haven, psych, reshape2, irr, ggplot2, ggpubr, effsize, emmeans, data.table, readr, Hmisc, RColorBrewer and tidyverse. We also thank the participants for their contribution to science; some of them may have experienced stress with the comprehension items.

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
