## [Reviewer comments · Royal Society Open Science]

Review History

RSOS-190876.R0 (Original submission)

Review form: Reviewer 1

Is the language acceptable?

Yes

Do you have any ethical concerns with this paper?

No

Have you any concerns about statistical analyses in this paper?

No

Recommendation?

Accept with minor revision

Comments to the Author(s)

The authors plan to test the efficacy of fact boxes in risk communication. They have proposed a large experiment in which participants will be randomly assigned to either receive information in

fact box format or receive information in paragraphs (in two different scenarios). I think the experiment is well designed, the power analysis and sample sizes are appropriate, and the proposed statistical analysis is sound. The proposed hypotheses are also quite reasonable. All elements of this preregistration are clear and the methodological detail is sufficient for future replication.

The research questions are important, though the tests are unlikely to shed light on the causes of the differences between fact box and paragraph presentation formats. Ideally I would prefer tests that are able to explain why fact boxes are better. This could be as simple as administering different types of objective comprehension questions and then testing for differences across types of research questions. This is not my research area so unfortunately I cannot provide additional recommendations here.

I would also prefer statistical tests that interact demographic variables or numeracy with the manipulation to test whether different risk communication strategies work better for different populations. This could be done as exploratory analysis -- it need not be taken into account in sample size calculations, but could be reported in the final paper.

The authors presented the PPS and lens for OAM generation, and implementation and measurement results are provided.

Review form: Reviewer 2

Is the language acceptable?

Yes

Do you have any ethical concerns with this paper?

No

Have you any concerns about statistical analyses in this paper?

No

Recommendation?

Accept with minor revision

Comments to the Author(s)

This is an excellent proposal: well-written, well-motivated. Relevant literature is cited, gap in research is identified, justification for this new data collection (instead of going for a meta-analysis) is very solid and convincing. Hypotheses all make sense. Similarly, methods make perfect sense, are described in sufficient detail, and allow for a test of the stated hypotheses. Power analysis (computed with G-Power) is standard and correct.

After all this praise, which is, basically, my green light for pursuing this work, here are some suggestions. Some concern the formulation of the materials, and they may come too late (I think this may also call for an editorial decision), as these materials have already been pre-tested. References are added in brackets – some of these brackets have a space upfront, some not. Looks a bit like coin-flipping which has and which has not.

Page 3, line 43: “diverse fields” ... Well, there are only two scenarios that have been used. “Diverse” somehow raises expectations – as if there will be a long list.

One scenario is preventative, the other is therapeutic. I would always mention these two in the same order (in intro, methods, figures, results). This is currently not done.

In table 1, there is not only a contrast between preventative and therapeutic, but also between children and adults. This is fine, such a confounding of age and disease is not problematic. But what I find disturbing is the fact that in Figure 1, where the information is displayed in a table, that is, in a fact box, the word “patient” is used. So you have text-only compared to fact-boxes,

the former use children, but in the latter they are no longer (neutral) children, but patients. I find this confounding completely unnecessary – after all, it amounts to varying something within your experimental manipulation.

Similarly, the older adults who are mentioned in the text-only condition DEVELOPPED influenza, while the fact-boxes speak of adults who SUFFERED from influenza. In my view, adding such differences to your treatment variable is not only unnecessary, it bad from a methodological point of view. In general: match the experimental conditions as closely as possible!

Page 13, line 35: "... a relative ..." (singular) "...them..." (plural). Is this okay? (English is not my native language)

Decision letter (RSOS-190876.R0)

24-Jun-2019

Dear Dr Brick

On behalf of the Editors, I am pleased to inform you that your Manuscript RSOS-190876 entitled "Risk communication in tables vs. text: a Registered Report randomised trial on 'fact boxes'" deemed suitable for in-principle acceptance in Royal Society Open Science subject to minor revision in accordance with the referee and editor suggestions. Please find their comments at the end of this email.

The reviewers and handling editors have recommended publication, but also suggest some minor revisions to your manuscript. Therefore, I invite you to respond to the comments and revise your manuscript.

Full author guidelines can be found here

<http://rsos.royalsocietypublishing.org/content/registered-reports>.

Kind regards

Professor Chris Chambers

Registered Reports Editor

Editor Comments to Author (Professor Chris Chambers):

Two expert reviewers have now assessed the manuscript. The assessments are overall positive but also note areas that will require clarification to achieve IPA. Reviewer 2, especially, raises concerns about potential design confounds or limitations that will need to be thoroughly addressed, either through revision or rebuttal. Please note that the suggested additional analyses by Reviewer 1 could be introduced at Stage 2 and need not be proposed in the Stage 1 manuscript.

Reviewer comments to Author:

Reviewer: 1

The authors plan to test the efficacy of fact boxes in risk communication. They have proposed a large experiment in which participants will be randomly assigned to either receive information in fact box format or receive information in paragraphs (in two different scenarios). I think the experiment is well designed, the power analysis and sample sizes are appropriate, and the proposed statistical analysis is sound. The proposed hypotheses are also quite reasonable. All elements of this preregistration are clear and the methodological detail is sufficient for future replication.

The research questions are important, though the tests are unlikely to shed light on the causes of the differences between fact box and paragraph presentation formats. Ideally I would prefer tests that are able to explain why fact boxes are better. This could be as simple as administering different types of objective comprehension questions and then testing for differences across types of research questions. This is not my research area so unfortunately I cannot provide additional recommendations here.

I would also prefer statistical tests that interact demographic variables or numeracy with the manipulation to test whether different risk communication strategies work better for different populations. This could be done as exploratory analysis -- it need not be taken into account in sample size calculations, but could be reported in the final paper.

Reviewer: 2

Comments to the Author(s)

This is an excellent proposal: well-written, well-motivated. Relevant literature is cited, gap in research is identified, justification for this new data collection (instead of going for a meta-analysis) is very solid and convincing. Hypotheses all make sense. Similarly, methods make perfect sense, are described in sufficient detail, and allow for a test of the stated hypotheses. Power analysis (computed with G-Power) is standard and correct.

After all this praise, which is, basically, my green light for pursuing this work, here are some suggestions. Some concern the formulation of the materials, and they may come too late (I think this may also call for an editorial decision), as these materials have already been pre-tested. References are added in brackets – some of these brackets have a space upfront, some not. Looks a bit like coin-flipping which has and which has not.

Page 3, line 43: “diverse fields” ... Well, there are only two scenarios that have been used. “Diverse” somehow raises expectations – as if there will be a long list.

One scenario is preventative, the other is therapeutic. I would always mention these two in the same order (in intro, methods, figures, results). This is currently not done.

In table 1, there is not only a contrast between preventative and therapeutic, but also between children and adults. This is fine, such a confounding of age and disease is not problematic. But what I find disturbing is the fact that in Figure 1, where the information is displayed in a table, that is, in a fact box, the word “patient” is used. So you have text-only compared to fact-boxes, the former use children, but in the latter they are no longer (neutral) children, but patients. I find

this confounding completely unnecessary – after all, it amounts to varying something within your experimental manipulation.

Similarly, the older adults who are mentioned in the text-only condition DEVELOPPED influenza, while the fact-boxes speak of adults who SUFFERED from influenza. In my view, adding such differences to your treatment variable is not only unnecessary, it bad from a methodological point of view. In general: match the experimental conditions as closely as possible!

Page 13, line 35: "... a relative ..." (singular) "...them..." (plural). Is this okay? (English is not my native language)

Author's Response to Decision Letter for (RSOS-190876.R0)

See Appendix A.

Decision letter (RSOS-190876.R1)

19-Jul-2019

Dear Dr Brick

On behalf of the Editor, I am pleased to inform you that your Manuscript RSOS-190876.R1 entitled "Risk communication in tables vs. text: a Registered Report randomised trial on 'fact boxes'" has been accepted in principle for publication in Royal Society Open Science.

You may now progress to Stage 2 and complete the study as approved. Before commencing data collection we ask that you:

- 1) Update the journal office as to the anticipated completion date of your study.
- 2) Register your approved protocol on the Open Science Framework (<https://osf.io/rr>) or other recognised repository, either publicly or privately under embargo until submission of the Stage 2 manuscript. Please note that a time-stamped, independent registration of the protocol is mandatory under journal policy, and manuscripts that do not conform to this requirement cannot be considered at Stage 2. The protocol should be registered unchanged from its current approved state, with the time-stamp preceding implementation of the approved study design.

Following completion of your study, we invite you to resubmit your paper for peer review as a Stage 2 Registered Report. Please note that your manuscript can still be rejected for publication at Stage 2 if the Editors consider any of the following conditions to be met:

- The results were unable to test the authors' proposed hypotheses by failing to meet the approved outcome-neutral criteria.
- The authors altered the Introduction, rationale, or hypotheses, as approved in the Stage 1 submission.
- The authors failed to adhere closely to the registered experimental procedures. Please note that any deviations from the approved experimental procedures must be communicated to the editor immediately for approval, and prior to the completion of data collection. Failure to do so can

result in revocation of in-principle acceptance and rejection at Stage 2 (see complete guidelines for further information).

- Any post-hoc (unregistered) analyses were either unjustified, insufficiently caveated, or overly dominant in shaping the authors' conclusions.
- The authors' conclusions were not justified given the data obtained.

We encourage you to read the complete guidelines for authors concerning Stage 2 submissions at <http://rsos.royalsocietypublishing.org/content/registered-reports>. Please especially note the requirements for data sharing, reporting the URL of the independently registered protocol, and that withdrawing your manuscript will result in publication of a Withdrawn Registration.

Once again, thank you for submitting your manuscript to Royal Society Open Science and we look forward to receiving your Stage 2 submission. If you have any questions at all, please do not hesitate to get in touch. We look forward to hearing from you shortly with the anticipated submission date for your stage two manuscript.

Kind regards,

Alice Power
Editorial Coordinator
Royal Society Open Science
openscience@royalsociety.org

on behalf of Professor Chris Chambers (Registered Reports Editor, Royal Society Open Science)
openscience@royalsociety.org

Author's Response to Decision Letter for (RSOS-190876.R1)

See Appendix B.

RSOS-190876.R2 (Revision)

Review form: Reviewer 1

Is the manuscript scientifically sound in its present form?

Yes

Are the interpretations and conclusions justified by the results?

Yes

Is the language acceptable?

Yes

Do you have any ethical concerns with this paper?

No

Have you any concerns about statistical analyses in this paper?

No

Recommendation?

Accept with minor revision

Comments to the Author(s)

The authors have appropriately implemented their preregistered analysis. The writing is clear and the data visualization is suitable. As expected, the data are able to test the authors' proposed hypotheses by passing the approved outcome-neutral criteria; the introduction, rationale and stated hypotheses are the same as the approved Stage 1 submission; the authors have adhered precisely to the registered experimental procedures; the exploratory analysis are justified and informative; and the conclusions are justified given the data

One final minor point: The recall effects are very small (29.8% vs. 27.7% across the conditions). The results abstract could be edited to reflect this small effect, e.g. by stating "Fact boxes – simple tabular messages – led to more comprehension ($d = .39$) and **slightly** more knowledge recall after six weeks ($d = .12$)". The discussion could also make the smallness of this effect clear. To what extent is 29.8% vs. 27.7% a relevant difference in real world settings?

Review form: Reviewer 2

Is the manuscript scientifically sound in its present form?

Yes

Are the interpretations and conclusions justified by the results?

Yes

Is the language acceptable?

Yes

Do you have any ethical concerns with this paper?

No

Have you any concerns about statistical analyses in this paper?

No

Recommendation?

Accept with minor revision

Comments to the Author(s)

I was also a reviewer of the initial proposal. I think the data are able to test the authors' proposed hypotheses by passing the approved outcome-neutral criteria (such as absence of floor and ceiling effects or success of positive controls).

As far as i can see, the Introduction, rationale and stated hypotheses are the same as the approved Stage 1 submission.

Moreover, i confirm that the authors adhered precisely to the registered experimental procedures. And yes, the authors' conclusions are justified given the data.

here are a few minor issues.

page 5/line 54. shouldnt it be "patients" (plural)?

page 8/ line 49ff: ... was were, but then on next page two times "will". Some leftovers from the preregistration and forgotten to update?

same on p10/40: ...responded see answered....

the two groups of bars in Figure 5 are indistinguishable in my black/white printout-

Figs 5 and 6: do the 95% CIs refer to the standard errors or to the standard deviations? I assume the former, but maybe better say so explicitly.

I found it a bit funny to see on page 25/line 26 that fact boxes led to more recall (based on a difference between 29.8 and 27.7, that is 2.1 percentage points, but then on page 31/17, a difference of 1.8 percentage points was described as "similar"

on page 38/19: I think it should read user-centered

page 37, top: there is a study by Garcia-Retamero & Hoffrage, 2013, SocSciMed, that looks into this: natural frequencies alone vs displayed jointly with visual aids - just in case the authors find it interesting in this context.

Decision letter (RSOS-190876.R2)

21-Feb-2020

Dear Dr Brick:

On behalf of the Editor, I am pleased to inform you that your Stage 2 Registered Report RSOS-190876.R2 entitled "Risk communication in tables vs. text: a Registered Report randomised trial on 'fact boxes'" has been deemed suitable for publication in Royal Society Open Science subject to minor revision in accordance with the referee suggestions. Please find the referees' comments at the end of this email.

The reviewers and Subject Editor have recommended publication, but also suggest some minor revisions to your manuscript. Therefore, I invite you to respond to the comments and revise your manuscript.

Please also ensure that all the below editorial sections are included where appropriate -- if any section is not applicable to your manuscript, please can we ask you to nevertheless include the heading, but explicitly state that the heading is inapplicable. An example of these sections is attached with this email.

- Ethics statement

- Data accessibility

It is a condition of publication that all supporting data are made available either as supplementary information or preferably in a suitable permanent repository. The data accessibility section should state where the article's supporting data can be accessed. This section

should also include details, where possible of where to access other relevant research materials such as statistical tools, protocols, software etc can be accessed. If the data has been deposited in an external repository this section should list the database, accession number and link to the DOI for all data from the article that has been made publicly available. Data sets that have been deposited in an external repository and have a DOI should also be appropriately cited in the manuscript and included in the reference list.

If you wish to submit your supporting data or code to Dryad (<http://datadryad.org/>), or modify your current submission to dryad, please use the following link:
[http://datadryad.org/submit?journalID=RSOS&manu=\(Document not available\)](http://datadryad.org/submit?journalID=RSOS&manu=(Document not available))

- **Competing interests**

- **Authors' contributions**

- **Acknowledgements**

- **Funding statement**

Because the schedule for publication is very tight, it is a condition of publication that you submit the revised version of your manuscript within 7 days (i.e. by the 29-Feb-2020). If you do not think you will be able to meet this date please let me know immediately.

Please note that Royal Society Open Science will introduce article processing charges for all new submissions received from 1 January 2018. Registered Reports submitted and accepted after this date will ONLY be subject to a charge if they subsequently progress to and are accepted as Stage 2 Registered Reports. If your manuscript is submitted and accepted for publication after 1 January 2018 (i.e. as a full Stage 2 Registered Report), you will be asked to pay the article processing charge, unless you request a waiver and this is approved by Royal Society Publishing. You can find out more about the charges at <https://royalsocietypublishing.org/rsos/charges>. Should you have any queries, please contact openscience@royalsociety.org.

Kind regards,

Anita Kristiansen
Editorial Coordinator

on behalf of Professor Chris Chambers
(Registered Reports Editor, Royal Society Open Science)
openscience@royalsociety.org

Associate Editor Comments to Author (Professor Chris Chambers):

Associate Editor: 1

Comments to the Author:

The two reviewers from Stage 1 have now assessed the Stage 2 manuscript. Both are satisfied with the submission and recommend publication following minor revisions to maximise transparency in the interpretation, consider additional literature in the Discussion, and fix some minor grammatical/clarity issues.

Comments to Author:
Reviewer: 1

Comments to the Author(s)

The authors have appropriately implemented their preregistered analysis. The writing is clear and the data visualization is suitable. As expected, the data are able to test the authors' proposed hypotheses by passing the approved outcome-neutral criteria; the introduction, rationale and stated hypotheses are the same as the approved Stage 1 submission; the authors have adhered precisely to the registered experimental procedures; the exploratory analysis are justified and informative; and the conclusions are justified given the data

One final minor point: The recall effects are very small (29.8% vs. 27.7% across the conditions). The results abstract could be edited to reflect this small effect, e.g. by stating "Fact boxes – simple tabular messages – led to more comprehension ($d = .39$) and **slightly** more knowledge recall after six weeks ($d = .12$)". The discussion could also make the smallness of this effect clear. To what extent is 29.8% vs. 27.7% a relevant difference in real world settings?

Reviewer: 2

Comments to the Author(s)

I was also a reviewer of the initial proposal. I think the data are able to test the authors' proposed hypotheses by passing the approved outcome-neutral criteria (such as absence of floor and ceiling effects or success of positive controls).

As far as i can see, the Introduction, rationale and stated hypotheses are the same as the approved Stage 1 submission.

Moreover, i confirm that the authors adhered precisely to the registered experimental procedures. And yes, the authors' conclusions are justified given the data.

here are a few minor issues.

page 5/line 54. shouldn't it be "patients" (plural)?

page 8/ line 49ff: ... was were, but then on next page two times "will". Some leftovers from the preregistration and forgotten to update?

same on p10/40: ...responded see answered....

the two groups of bars in Figure 5 are indistinguishable in my black/white printout-

Figs 5 and 6: do the 95% CIs refer to the standard errors or to the standard deviations? I assume the former, but maybe better say so explicitly.

I found it a bit funny to see on page 25/line 26 that fact boxes led to more recall (based on a difference between 29.8 and 27.7, that is 2.1 percentage points, but then on page 31/17, a difference of 1.8 percentage points was described as "similar"

on page 38/19: I think it should read user-centered

page 37, top: there is a study by Garcia-Retamero & Hoffrage, 2013, SocSciMed, that looks into this: natural frequencies alone vs displayed jointly with visual aids - just in case the authors find it interesting in this context.

Author's Response to Decision Letter for (RSOS-190876.R2)

See Appendix C.

Decision letter (RSOS-190876.R3)

27-Feb-2020

Dear Dr Brick,

It is a pleasure to accept your manuscript entitled "Risk communication in tables vs. text: a Registered Report randomised trial on 'fact boxes'" in its current form for publication in Royal Society Open Science.

on behalf of Prof Chris Chambers (Subject Editor)
openscience@royalsociety.org

Appendix A

Dear Editors and Reviewers: thank you for the opportunity to revise and resubmit this paper, *Risk communication in tables vs. text: a Registered Report randomised trial on 'fact boxes'*. We appreciate your thoughtful comments and enjoyed the process of revising the methods before data collection. Below, positive reviewer comments without suggestions are omitted. There were no changes to the title page, abstract, or keywords.

Reviewer 1

- 1) [...] The research questions are important, though the tests are unlikely to shed light on the causes of the differences between fact box and paragraph presentation formats. Ideally I would prefer tests that are able to explain why fact boxes are better. This could be as simple as administering different types of objective comprehension questions and then testing for differences across types of research questions. This is not my research area so unfortunately I cannot provide additional recommendations here.

Response: That would also be useful. The comprehension questions have some diversity in terms of asking for numeric values vs. some interpretation or understanding of how the numbers relate. We could include exploratory analyses between these item types at Stage 2. This is a bit looser and more speculative so we'll leave it out of Stage 1. If fact boxes are shown to be understood better, that would inform the design of studies directly investigating the mechanism, for example studies with more cognitive methods such as gaze, engagement, encoding, or recall vs. recognition.

- 2) I would also prefer statistical tests that interact demographic variables or numeracy with the manipulation to test whether different risk communication strategies work better for different populations. This could be done as exploratory analysis -- it need not be taken into account in sample size calculations, but could be reported in the final paper.

Response: We are also keenly interested in that. We didn't include any moderation by demographics in Stage 1 because the design is underpowered for moderation. We may revisit this topic for exploratory analyses in Stage 2.

Reviewer 2

- 1) [...] here are some suggestions. Some concern the formulation of the materials, and they may come too late (I think this may also call for an editorial decision), as these materials have

already been pre-tested. References are added in brackets – some of these brackets have a space upfront, some not. Looks a bit like coin- flipping which has and which has not.

Response: Despite the pre-testing, the editor and authors both support revising the materials before the main study. More revision details below. The extra spaces before any citations in the main text have been removed.

- 2) Page 3, line 43: “diverse fields” ... Well, there are only two scenarios that have been used. “Diverse” somehow raises expectations – as if there will be a long list. One scenario is preventative, the other is therapeutic. I would always mention these two in the same order (in intro, methods, figures, results). This is currently not done.

Response: We made these changes as suggested. "Diverse" was removed. The sentence now reads: "The results of the current study will support evidence-based communication in fields where harms and benefits need to be summarised for multiple options." Where the medical topics were mentioned or described, now they are explicitly labeled as "preventative and therapeutic" in all sections (pps. 1, 6). The topics were already explicitly labeled as such in the Methods under topic, and later where they are referenced in the Analytic Plan, we expect readers will refer to earlier sections.

- 3) In table 1, there is not only a contrast between preventative and therapeutic, but also between children and adults. This is fine, such a confounding of age and disease is not problematic. But what I find disturbing is the fact that in Figure 1, where the information is displayed in a table, that is, in a fact box, the word “patient” is used. So you have text-only compared to fact-boxes, the former use children, but in the latter they are no longer (neutral) children, but patients. I find this confounding completely unnecessary – after all, it amounts to varying something within your experimental manipulation.

Response: We agree. All four instances of "patients" were revised to "children" in the ear infection fact box (Figure 1). "Patients" did not occur anywhere else in the materials. This change were double-checked between the text boxes and control conditions, between topics, and against the comprehension questions.

- 4) Similarly, the older adults who are mentioned in the text-only condition DEVELOPPED influenza, while the fact-boxes speak of adults who SUFFERED from influenza. In my view, adding such differences to your treatment variable is not only unnecessary, it bad from a methodological point of view. In general: match the experimental conditions as closely as possible!

Response: Thanks for pointing out this confound. We agree about matching the conditions. Three instances of "suffer[]" were revised in the Influenza text-only control (changed to "developed", "did", and "experienced" in fitting with the other condition). "Suffer[]" also appeared in many comprehension items, and was revised in line with the new text.

In addition, in the ear infection text, "still [had pain]" was revised to remove the value statement "still".

Both these changes were double-checked between the text boxes and control conditions, between topics, and against the comprehension questions.

In addition, during this revision we noticed a potential confound with how the term "placebo" was introduced between the conditions. In the ear infection condition instructions, the term "placebo" was previously followed by "(sugar pill)" to help readers understand what placebo means. To make the influenza condition more similar, we now took "Older adults with placebo received an injection with a saline solution" and added "(no vaccine)". Similarly, in the text-only control of the influenza condition, we added "(no vaccine)" after the first mention of "placebo". Please see the new questionnaire for the full paragraphs, fact box images, and context.

- 5) Page 13, line 35: "... a relative ..." (singular) "...them..." (plural). Is this okay? (English is not my native language)

Response: For clarity, this sentence was revised to disambiguate who was recommending: "Participants will be prompted to imagine that a relative is making a decision on this subject and the participant will be asked whether they would recommend the treatment."

Appendix B

We confirm all of the conditions for the Stage 2 submission. Please see the cover letter and thank you.

Appendix C

Dear Editors and Reviewers: thank you for the opportunity to revise and resubmit this paper, *Risk communication in tables vs. text: a Registered Report randomised trial on 'fact boxes'*. Below, positive reviewer comments without suggestions are omitted.

Reviewer 1

- 1) One final minor point: The recall effects are very small (29.8% vs. 27.7% across the conditions). The results abstract could be edited to reflect this small effect, e.g. by stating “Fact boxes—simple tabular messages—led to more comprehension ($d = .39$) and **slightly** more knowledge recall after six weeks ($d = .12$)”. The discussion could also make the smallness of this effect clear. To what extent is 29.8% vs. 27.7% a relevant difference in real world settings?

Response: We agree and made the following changes to be clear about the meaningfulness of this difference: 1) “slightly” was added to the abstract and Discussion; and 2) where this effect is presented in the Discussion it is now treated more circumspectly: “Fact boxes also led to slightly better recall (knowledge) after six weeks ($d = .12$; 29.8% vs. 27.7% for text alone). Higher scores are better, but it is unknown whether this size effect would lead to meaningful differences in informed decision making.”

Reviewer 2

- 1) page 5/line 54. shouldn't it be "patients" (plural)?

Response: Corrected.

- 2) page 8/ line 49ff: ... was were, but then on next page two times "will". Some leftovers from the preregistration and forgotten to update?

Response: The ‘Current study’ and ‘Analytic plan’ passages were changed to consistently use past tense.

- 3) same on p10/40: ...responded see answered....

Response: Corrected, along with a few other minor tense errors in ‘Procedure and open data, code, materials’.

- 4) the two groups of bars in Figure 5 are indistinguishable in my black/white printout-

Response: The color palette of Figure 5 was changed to the same as Figure 6. It affords more contrast between the condition hue.

- 5) Figs 5 and 6: do the 95% CIs refer to the standard errors or to the standard deviations? I assume the former, but maybe better say so explicitly.

Response: Both figure captions were edited to read: "Error bars are 95% CIs based on standard errors."

- 6) I found it a bit funny to see on page 25/line 26 that fact boxes led to more recall (based on a difference between 29.8 and 27.7, that is 2.1 percentage points, but then on page 31/17, a difference of 1.8 percentage points was described as "similar"

Response: Yes, we agree. Based on feedback from Reviewer 1, we made a set of changes that downplay the importance of the recall effect size (please see above). The passage in 'Attrition' was changed to: "Participation in the follow-up survey appeared slightly higher for participants who saw a fact box (73.2%) rather than text only (71.4%)." This is more consistent with how effects of that size are described verbally in the rest of the manuscript.

- 7) on page 38/19: I think it should read user-centered

Response: "user-centred" is the British spelling.

- 8) page 37, top: there is a study by Garcia-Retamero & Hoffrage, 2013, SocSciMed, that looks into this: natural frequencies alone vs displayed jointly with visual aids - just in case the authors find it interesting in this context.

Response: We looked again at this passage. Based on the existing paragraphs and references, this paper is definitely relevant, but perhaps not necessary for inclusion. We decided not to add this reference, but we do appreciate the suggestion to make sure we were aware of it.

Thank you to the editor and reviewers. This whole Registered Report process has been a pleasure.

-The Authors